# STATISTICAL TEST FOR ANOMALY DETECTIONS USING VARIATIONAL AUTO-ENCODERS BY SELECTIVE INFERENCE

## ABSTRACT

Over the past decade, Variational Autoencoders (VAE) have become a widely used tool for anomaly detection (AD), with research advancing from algorithm development to real-world applications. However, a critical challenge remains—the lack of a reliable method to rigorously assess the reliability of detected anomalies, which restricts its use in high-stakes decision-making tasks such as medical diagnostics. To overcome this limitation, we introduce the VAE-AD Test, a novel approach for quantifying the statistical reliability of VAE-based AD. The key advantage of the VAE-AD Test lies in its ability to properly control the probability of misidentifying anomalies under a pre-specified level of guarantee $\alpha$ (e.g., 0.05). Specifically, by carefully analyzing the AD process of VAE, which operates through piecewise-linear functions, and leveraging the Selective Inference (SI) framework to assign valid p-values to the detected anomalies, we prove that theoretical control of the false detection rate is achievable. Experiments conducted on both synthetic and real-world datasets robustly support our theoretical results, showcasing the VAE-AD Test's superior performance. To our knowledge, this is the first work capable of conducting valid statistical inference to assess the reliability of VAE-based AD.

## 1 INTRODUCTION

Anomaly detection (AD) is the process of identifying unusual deviations in data that do not conform to expected behavior. AD is crucial across various domains because it provides early warnings of potential issues, thereby enabling timely interventions to prevent critical events. Traditional AD techniques, while effective in simple scenarios, frequently fall short when dealing with complex data, thus motivating the use of deep learning-based AD to better handle such complexities. In this study, we focus on AD using the Variational Auto-Encoder (VAE), and its application to medical images. In the training phase of VAE-based AD, the VAE learns the distribution of normal images by training exclusively on images that do not contain abnormal regions. The parameters of a VAE are optimized to minimize the reconstruction error, thereby learning a compressed representation of the normal data. In the test phase, when a test image is fed into the trained VAE, the model attempts to reconstruct the image based on its learned representation. Since the VAE is trained on normal data, it would successfully reconstruct the normal regions of the image, while it would fail to properly reconstruct the abnormal regions that were not included in the normal data. Therefore, regions with large reconstruction errors are detected as abnormal regions. Figure 1 shows an example of VAE-based AD for brain tumor images.

When VAE-based AD is employed for high-stakes decision-making tasks, such as medical diagnosis, there is a significant risk that model inaccuracies might lead to critical errors, potentially resulting in false detections. To address this issue, we develop a statistical test for VAE-based AD, which we call *VAE-AD Test*. The proposed VAE-AD test enables us to obtain a quantifiable and interpretable measure for the detected anomaly region in the form of $p$-value. The obtained $p$-value represents the probability that the detected anomaly regions are obtained by chance due to the randomness contained in the data. It is important to note that the statistical test for detected abnormal regions is considered as a *data-driven hypothesis*, as the abnormal region is selected based on the test image itself. In other words, since both of the selection of the hypothesis (selection of abnormal regions) and the evaluation of the hypothesis (evaluation of abnormal regions) are performed on the same data,

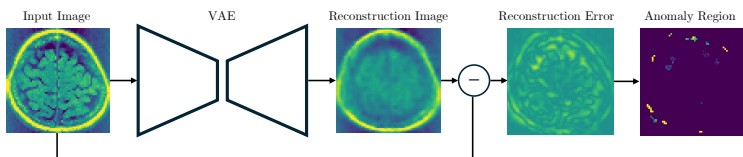

(a) Image without tumor region. $p_{\text{naive}} = 0.000$ (false detection) and $p_{\text{selective}} = 0.668$ (true negative).

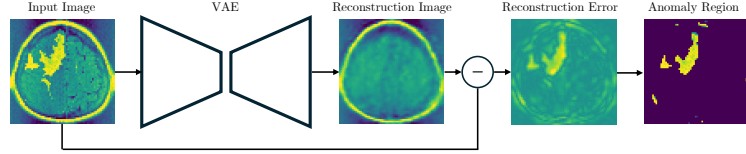

(b) Image with tumor region. $p_{\text{naive}} = 0.000$ (true detection) and $p_{\text{selective}} = 0.000$ (true detection).

Figure 1: An illustration of the proposed VAE-AD Test in brain image analysis. When an anomaly region is detected based on the difference between the original and the reconstructed images by a VAE, the VAE-AD Test provides a $p$-value to quantify its statistical reliability. The upper plot shows the results of the VAE-AD Test and the conventional method, the latter of which does not consider the fact that the anomaly region is detected by VAE. The lower plot shows the results for a case with anomaly regions. With the proposed method ($p_{\text{selective}}$), correct decisions are made in both cases; the former has a large $p$-value and the latter has a small $p$-value. In contrast, with the conventional method ($p_{\text{naive}}$), both $p$-values are small, indicating false detection in the former case.

applying traditional statistical test to the selected hypothesis leads to selection bias. Therefore, in this study, we introduce the *Conditional Selective Inference (CSI)* framework to remove the selection bias.

**Related Works.** Over the last decade, there has been a significant pursuit in applying deep learning techniques to AD problems (Chalapathy & Chawla, 2019; Pang et al., 2021; Tao et al., 2022). A large number of studies have been conducted for unsupervised AD using VAEs (Baur et al., 2021; Chen & Konukoglu, 2018; Chow et al., 2020; Jana et al., 2022). In this study, we focus on the task of identifying the anomalous regions in the input image, which is called *anomaly localization* within the AD tasks (Zimmerer et al., 2019; Lu & Xu, 2018; Baur et al., 2019).

There are mainly two research directions for improving VAEs for AD. The first direction is on improving the detection rate (Zimmerer et al., 2019; Dehaene et al., 2020), while the second direction is on modifying the VAE itself to make it suitable for AD (Baur et al., 2019; Chen & Konukoglu, 2018; Wang et al., 2020). However, to our knowledge, there has been no existing studies for quantifying the statistical reliability of detected abnormal regions with theoretical validity. In traditional statistical tests, the hypothesis needs to be predetermined and must remain independent of the data. However, in data-driven approaches, it is necessary to select hypotheses based on the data and then assess the reliability of the hypotheses using the same data. This issue, known as *double dipping*, arises because the same data is used for both the selection and evaluation of hypotheses, leading to selection bias (Breiman, 1992). Because anomalies are detected based on data (a test image), when evaluating the reliability of the detected anomalies using the same data, the issue of selection bias arises.

CSI has recently gained attention as a framework for statistical hypothesis testing of data-driven hypotheses (Lee et al., 2016; Taylor & Tibshirani, 2015). CSI was initially developed for the statistical inference of feature selection in linear models (Fithian et al., 2015; Tibshirani et al., 2016; Loftus & Taylor, 2014; Suzumura et al., 2017; Le Duy & Takeuchi, 2021; Sugiyama et al., 2021; Duy & Takeuchi, 2022), then extended to various problems (Lee et al., 2015; Choi et al., 2017; Chen & Bien, 2020; Tanizaki et al., 2020; Duy et al., 2020; Gao et al., 2022; Le Duy et al., 2024), and later to neural networks (Duy et al., 2022; Miwa et al., 2023; Shiraishi et al., 2024; Katsuoka et al., 2024), but none of these studies focused on inference on VAE.

**Contributions.** To our knowledge, this is the first formulation of an approach that provides a quantifiable and interpretable measure for the reliability of VAE-based AD, presented in the form of a $p$-value within a statistical testing framework. The second contribution is the development of an SI method for VAEs, which entails characterizing the hypothesis selection event by a VAE. Finally, our third contribution is demonstrating the effectiveness of the proposed VAE-AD Test through numerical experiments with synthetic data and brain tumor images.

## 2 ANOMALY DETECTION (AD) BY VAE

**Variational Autoencoder (VAE).** VAEs are generative models consisting of an encoder network and a decoder network Kingma & Welling (2013). Given an input image (denoted by $x \in \mathbb{R}^n$), it is encoded as a latent vector (denoted by $z \in \mathbb{R}^m$), and the latent vector is decoded back to the input image, where $n$ is the number of pixels of an image and $m$ is the dimension of a latent vector. In the generative process, it is assumed that a latent vector $z$ is sampled from a prior distribution $p_{\theta*}(z)$ and then, image $x$ is sampled from a conditional distribution $p_{\theta*}(x|z)$. The prior distribution $p_{\theta*}(z)$ and the conditional distribution $p_{\theta*}(x|z)$ belongs to family of distributions parametrized by $\theta$ and $\theta*$ denotes the true value of the parameter. The encoder network approximates the posterior distribution $p_\theta(z|x)$ by the parametric distribution $q_\phi(z|x)$, where $\phi$ represents the set of parameters, while the decoder network estimates the conditional distribution by $p_\theta(x|z)$. The encoder and the decoder networks of a VAE are trained by maximizing so-called evidence lower bound (ELBO): $L_{\theta,\phi} = \mathbb{E}_{q_\theta(z|x)} [\log p_\phi(x|z)] - \mathrm{KL} [q_\theta(z|x)||p(z)]$, where $\mathrm{KL}[\cdot||\cdot]$ is the Kullback-Leibler divergence between two distributions. We model the approximated posterior distribution $q_\phi(z|x)$ as a normal distribution $N(\mu_\phi(x), I_n\sigma_\phi^2(x))$, where $\mu_\phi(x)$ and $\sigma_\phi^2(x)$ are the outputs of the encoder network. The conditional distribution $p_\theta(x|z)$ is also modeled as a normal distribution $N(\mu_\theta(z), I_n)$, where $\mu_\theta(z)$ is the output of the decoder network. Furthermore, the prior distribution $p_{\theta*}(z)$ is modeld as a standard normal distribution $N(0, I_m)$. The structure of the VAE used in this study is shown in Appendix A.1.

**Anomaly Detection Using VAEs.** VAEs can be effectively used for anomaly localization task. The goal of anomaly localization is to identify the abnormal region within a given test image. In the training phase, we assume that only normal images (e.g., brain images without tumors) are available. A VAE is trained on normal images to learn a compact representation of the normal image distribution in the latent space. In the test phase, a test image $x$ is fed into the trained VAE, and a reconstructed image is obtained by using the encoder and the decoder as $\hat{x} = \mu_\theta(\mu_\phi(x))$. Since the VAE is trained only on normal images, normal region in the test image would be reconstructed well, whereas the reconstruction error of abnormal regions would be high. Therefore, it is reasonable to define the degree of anomaly of each pixel as

$$E_i(x) = |x_i - \hat{x}_i|, i \in [n], \tag{1}$$

where $x_i$ and $\hat{x}_i$ is the $i^{\text{th}}$ pixel value of $x$ and $\hat{x}$, respectively. Using a user-specified threshold $\lambda > 0$, the anomaly region of a test image $x$ is defined as

$$A_x = \{i \in |n| \mid E_i(x) \geq \lambda\}. \tag{2}$$

As for the definition of the anomaly region, there are possibilities other than those given by Eqs. (1) and (2). In this paper, we proceed with these choices, but the proposed VAE-AD Test is generally applicable to other choices.

## 3 STATISTICAL TEST FOR ABNORMAL REGIONS

**Statistical model of an image.** To formulate the reliability assessment of the abnormal region as a statistical testing problem, it is necessary to introduce a statistical model of an image. In this study, an image is considered as a sum of true signal component $s \in \mathbb{R}^n$ and noise component $\epsilon \in \mathbb{R}^n$. Regarding the true signal component, each pixel can have an arbitrary true signal value without any particular assumption or constraint. On the other hand, regarding the noise component, it is assumed to follow a normal distribution, and their covariance matrix is estimated using normal data different from that used for the training of the VAE. Namely, an image with $n$ pixels can be represented as an $n$-dimensional random vector

$$X = (X_1, \ldots, X_n) = s + \epsilon, \; \epsilon \sim N(\mathbf{0}, \Sigma), \tag{3}$$

where $s \in \mathbb{R}^n$ is the true signal vectors, and $\epsilon \in \mathbb{R}^n$ is the noise vector with covariance matrix $\Sigma$. In the following, the capital $X$ denotes an image as a random vector, while the lowercase $x$ represents an observed image. To formulate the statistical test, we consider the AD using VAEs in Eq. equation 2 as a function $\mathcal{A}$ that maps a random input image $X$ to the abnormal region $A_X$, i.e.,

$$\mathcal{A} : \mathbb{R}^n \ni X \mapsto A_X \in 2^{[n]}, \tag{4}$$

where $2^{[n]}$ is the power set of $[n] := \{1, 2, \ldots, n\}$.

**Formulation of statistical test.** Our goal is to make a judgment whether the abnormal region $A_{\boldsymbol{X}}$ merely appears abnormal due to the influence of random noise, or if there is a true anomaly in the true signal in the abnormal region. In order to quantify the reliability of the detected abnormal region, the statistical test is performed for the difference between the true signal in the abnormal region $\{s_i\}_{i \in A_{\boldsymbol{X}}}$ and the true signal in the normal region $\{s_i\}_{i \in A_{\boldsymbol{X}}^c}$ where $A_{\boldsymbol{X}}^c$ is the complement of the abnormal region. In this study, as an example, we consider the hypothesis for the difference in true mean signals between $A_{\boldsymbol{X}}$ and $A_{\boldsymbol{X}}^c$ by considering the following null and alternative hypotheses:

$$\mathrm{H}_0 : \frac{1}{|A_{\boldsymbol{X}}|} \sum_{i \in A_{\boldsymbol{X}}} s_i = \frac{1}{|A_{\boldsymbol{X}}^c|} \sum_{i \in A_{\boldsymbol{X}}^c} s_i, \text{ v.s. } \mathrm{H}_1 : \frac{1}{|A_{\boldsymbol{X}}|} \sum_{i \in A_{\boldsymbol{X}}} s_i \neq \frac{1}{|A_{\boldsymbol{X}}^c|} \sum_{i \in A_{\boldsymbol{X}}^c} s_i. \tag{5}$$

For clarity, we mainly consider a test for the mean difference as a specific example — however, the proposed VAE-AD Test is applicable to a more general class of statistical tests. Specifically, let $\boldsymbol{\eta} \in \mathbb{R}^n$ be an arbitrary $n$-dimensional vector depending on the abnormal region $A_{\boldsymbol{X}}$. Then, the proposed method can cover a statistical test represented as

$$\mathrm{H}_0 : \boldsymbol{\eta}^\top \boldsymbol{s} = c \quad \text{v.s.} \quad \mathrm{H}_1 : \boldsymbol{\eta}^\top \boldsymbol{s} \neq c, \tag{6}$$

where $c$ is an arbitrary constant. The formulation in Eq. (6) covers a wide range of practically useful statistical tests. In fact, Eq. (5) is a special case of Eq. (6). It can cover differences not only in means but also in other measures such as maximum difference, and differences after applying some image filters (e.g., Gaussian filter).

**Test statistic.** To evaluate the hypothesis defined in Eq. equation 5, we define the test statistic as

$$T(\boldsymbol{X}) = \frac{1}{|A_{\boldsymbol{X}}|} \sum_{i \in A_{\boldsymbol{X}}} X_i - \frac{1}{|A_{\boldsymbol{X}}^c|} \sum_{i \in A_{\boldsymbol{X}}^c} X_i = \boldsymbol{\eta}^\top \boldsymbol{X}, \tag{7}$$

where $\boldsymbol{\eta} = \frac{1}{|A_{\boldsymbol{X}}|} \mathbf{1}_{A_{\boldsymbol{X}}} - \frac{1}{|A_{\boldsymbol{X}}^c|} \mathbf{1}_{A_{\boldsymbol{X}}^c}$ and $\mathbf{1}_A \in \mathbb{R}^n$ is a vector with 1 if $i \in A_{\boldsymbol{X}}$ and 0 otherwise.

**Naive $p$-values.** When the test statistic in Eq. (7) is used for the statistical test in Eq. (5), the $p$-value can be easily calculated *if $\boldsymbol{\eta}$ does not depend on the image $\boldsymbol{X}$, i.e., if the abnormal region $A_{\boldsymbol{X}}$ is detected without looking at the $\boldsymbol{X}$*. In this *unrealistic* situation, the $p$-value, which we call *naive p-value* can be computed as $p_{\mathrm{naive}} = \mathbb{P}_{\mathrm{H}_0}(|T(\boldsymbol{X})| \geq |T(\boldsymbol{x})|)$, where $\boldsymbol{X}$ is a random vector and $\boldsymbol{x}$ is the observed image. Under the unrealistic assumption, the $p_{\mathrm{naive}}$ can be easily computed because the null distribution of $T(\boldsymbol{X}) = \boldsymbol{\eta}^\top \boldsymbol{X}$ is normally distributed with $N(0, \boldsymbol{\eta}^\top \Sigma \boldsymbol{\eta})$. Unfortunately, however, in the actual situation where $\boldsymbol{\eta}$ depends on $\boldsymbol{X}$, a statistical test using $p_{\mathrm{naive}}$ is *invalid* in the sense that $P_{\mathrm{H}_0}(p_{\mathrm{naive}} \leq \alpha) > \alpha, \exists \alpha \in [0, 1]$. Namely, the probability of Type I error (an error that a normal region is mistakenly detected as anomaly) cannot be controlled at the desired level $\alpha$.

## 4 CONDITIONAL SELECTIVE INFERENCE (CSI) FOR VAE-BASED AD

In this section, we present the proposed VAE-AD Test, a valid statistical test for VAE-based AD task.

### 4.1 CONDTIONAL SELECTIVE INFERENCE (CSI)

In CSI, $p$-values are computed based on the null distribution conditional on a event that a certain hypothesis is selected. The goal of CSI is to compute a $p$-value that satisfies

$$P_{\mathrm{H}_0}(p \leq \alpha \mid A_{\boldsymbol{X}} = A) \leq \alpha, \tag{8}$$

where the condition part $A_{\boldsymbol{X}} = A$ in Eq. (8) indicates that we only consider images $\boldsymbol{X}$ for which a certain hypothesis (abnormal region) $A$ is detected. If the conditional type I error can be controlled as in Eq. (8) for all possible hypotheses $A \in 2^{[n]}$, then, by the law of total probability, the marginal type I error can also be controlled for all $\alpha \in (0, 1)$ because

$$P_{\mathrm{H}_0}(p \leq \alpha) = \sum_{A \in 2^{[n]}} P_{\mathrm{H}_0}(A)(p \leq \alpha \mid A_{\boldsymbol{X}} = A) \leq \alpha. \tag{10}$$

Therefore, in order to perform valid statistical test, we can employ $p$-values conditional on the hypothesis selection event. To compute a $p$-value that satisfies Eq. (8), we need to derive the sampling distribution of the test-statistic

$$T(\boldsymbol{X}) | \{A_{\boldsymbol{X}} = A_{\boldsymbol{x}}\}. \tag{9}$$

## 4.2 CSI FOR PIECEWISE-ASSIGNMENT FUNCTIONS

We derive the CSI for algorithms expressed in the form of a *piecewise-assignment function*. Later on, we show that the mapping $\mathcal{A} : \boldsymbol{X} \mapsto A_{\boldsymbol{X}}$ in Eq. (4) is a piecewise-assignment function, and this will result in the proposed VAE-AD Test.

**Definition 1** (Piecewise-Assignment Function)**.** Let us consider a function $M : \mathbb{R}^n \ni \boldsymbol{X} \mapsto M_{\boldsymbol{X}} \in \mathcal{M}$ which assigns an image $\boldsymbol{X}$ to a hypothesis among a finite set of hypotheses $\mathcal{M}$. We call the function $M$ a piecewise-assignment function if it is written as

$$
M_{\boldsymbol{X}} = \begin{cases} M_1, & \text{if} \quad \boldsymbol{X} \in \mathcal{P}_1^M, \\ \vdots \\ M_k, & \text{if} \quad \boldsymbol{X} \in \mathcal{P}_k^M, \\ \vdots \\ M_{K^M}, & \text{if} \quad \boldsymbol{X} \in \mathcal{P}_{K^M}^M, \end{cases} \tag{10}
$$

where $\mathcal{P}_k^M$, $k \in [K^M]$, represents a polytope in $\mathbb{R}^n$ which can be written as $\mathcal{P}_k^M = \{\boldsymbol{X} \in \mathbb{R}^n \mid \boldsymbol{\Delta}_k^M \boldsymbol{X}' \leq \boldsymbol{\delta}_k^M\}$ using a certain matrix $\boldsymbol{\Delta}_k^M$ and a vector $\boldsymbol{\delta}_k^M$ with appropriate sizes, and $K^M$ is the number of polytopes. Here, we note that the same hypothesis may be assigned to different polytopes.

When a hypothesis is selected by a piecewise-assignment function in the form of Eq. (10), the following theorem tells that the conditional $p$-value that satisfies Eq. (8) can be derived by using truncated normal distribution.

**Theorem 1.** *Consider a random image $\boldsymbol{X}$ and an observed image $\boldsymbol{x}$. Let $M_{\boldsymbol{X}}$ and $M_{\boldsymbol{x}}$ be the hypotheses obtained by applying a piecewise-assignment function in the form of Eq. (10) to $\boldsymbol{X}$ and $\boldsymbol{x}$, respectively. Let $\boldsymbol{\eta} \in \mathbb{R}^n$ be a vector depending on $M_{\boldsymbol{x}}$, and consider a test statistic in the form of $T(\boldsymbol{X}) = \boldsymbol{\eta}^\top \boldsymbol{X}$. Furthermore, define*

$$
\mathcal{Q}_{\boldsymbol{X}} = \left( I_n - \frac{\Sigma \boldsymbol{\eta} \boldsymbol{\eta}^\top}{\boldsymbol{\eta}^\top \Sigma \boldsymbol{\eta}} \right) \boldsymbol{X} \text{ and } \mathcal{Q}_{\boldsymbol{x}} = \left( I_n - \frac{\Sigma \boldsymbol{\eta} \boldsymbol{\eta}^\top}{\boldsymbol{\eta}^\top \Sigma \boldsymbol{\eta}} \right) \boldsymbol{x}.
$$

*Then, the conditional distribution*

$$
T(\boldsymbol{X}) \mid \{M_{\boldsymbol{X}} = M_{\boldsymbol{x}}, \mathcal{Q}_{\boldsymbol{X}} = \mathcal{Q}_{\boldsymbol{x}}\}
$$

*is a truncated normal distribution $TN(\boldsymbol{\eta}^\top \boldsymbol{\mu}, \boldsymbol{\eta}^\top \Sigma \boldsymbol{\eta}; \mathcal{Z})$ with the mean $\boldsymbol{\eta}^\top \boldsymbol{\mu}$, the variance $\boldsymbol{\eta}^\top \Sigma \boldsymbol{\eta}$, and the truncation intervals $\mathcal{Z}$. The truncation intervals $\mathcal{Z}$ is represented as*

$$
\mathcal{Z} = \bigcup_{k:M_k = M_{\boldsymbol{x}}} [L_k^M, U_k^M],
$$

*where, for $k \in [K^M]$, $L_k^M$ and $U_k^M$ are defined as follows:*

$$
L_k^M = \max_{j:(\boldsymbol{\beta}_k^M)_j > 0} \frac{(\boldsymbol{\alpha}_k^M)_j}{(\boldsymbol{\beta}_k^M)_j}, \ U_k^M = \min_{j:(\boldsymbol{\beta}_k^M)_j < 0} \frac{(\boldsymbol{\alpha}_k^M)_j}{(\boldsymbol{\beta}_k^M)_j}
$$

*with $\boldsymbol{\alpha}_k^M = \boldsymbol{\delta}_k^M - \boldsymbol{\Delta}_k^M \mathcal{Q}_{\boldsymbol{x}}$ and $\boldsymbol{\beta}_k^M = \boldsymbol{\Delta}_k^M \Sigma \boldsymbol{\eta} (\Sigma \boldsymbol{\eta}^\top \Sigma \boldsymbol{\eta})^{-1}$.*

The proof of Theorem 1 is deferred to Appendix A.2. Using the sampling distribution of the test statistic $T(\boldsymbol{X})$ conditional on $\{M_{\boldsymbol{X}} = M_{\boldsymbol{x}}, Q_{\boldsymbol{X}} = Q_{\boldsymbol{x}}\}$ in Theorem 1, we can define the $p$-value as

$$
p_{\text{selective}} = \mathbb{P}_{\text{H}_0}(|T(\boldsymbol{X})| \geq |T(\boldsymbol{x})| \mid M_{\boldsymbol{X}} = M_{\boldsymbol{x}}, \mathcal{Q}_{\boldsymbol{X}} = \mathcal{Q}_{\boldsymbol{x}}). \tag{11}
$$

The selective $p$-value $p_{\text{selective}}$ defined in Eq. equation 11 satisfies

$$
\mathbb{P}_{\text{H}_0}(p_{\text{selective}} \leq \alpha \mid M_{\boldsymbol{X}} = M_{\boldsymbol{x}}) = \alpha, \ \forall \alpha \in [0, 1]
$$

because $Q_{\boldsymbol{X}}$ is independent of the test statistic $T(\boldsymbol{X}) = \boldsymbol{\eta}^\top \boldsymbol{X}$. From the discussion in §4.1, a valid statistical test can be conducted by using $p_{\text{selective}}$ in Eq. (11).

## 4.3 PIECEWISE-LINEAR FUNCTIONS

We showed that, if the hypothesis selection algorithm is represented in the form of piecewise-assignment function, we can formulate valid selective $p$-values. The purpose of this subsection is to set the stage for demonstrating in the next subsection how the entire process of a trained VAE can be depicted as a *piecewise-linear function*, and how VAE-based AD algorithm in Eq. (4) is represented as a piecewise-assignment function.

**Definition 2** (Piecewise-Linear Function). A piecewise-linear function $f : \mathbb{R}^n \to \mathbb{R}^m$ is written as:

$$f(\boldsymbol{X}) = \begin{cases} \Psi_1^f \boldsymbol{X} + \boldsymbol{\psi}_1^f, & if \ \boldsymbol{X} \in \mathcal{P}_1^f, \\ \vdots \\ \Psi_k^f \boldsymbol{X} + \boldsymbol{\psi}_k^f, & if \ \boldsymbol{X} \in \mathcal{P}_k^f, \\ \vdots \\ \Psi_{K^f}^f \boldsymbol{X} + \boldsymbol{\psi}_{K^f}^f, & if \ \boldsymbol{X} \in \mathcal{P}_{K^f}^f, \end{cases} \tag{12}$$

where $\mathcal{P}_k^f$ represents a polytope in $\mathbb{R}^n$ written as $\mathcal{P}_k^f = \{\boldsymbol{X} \in \mathbb{R}^n \mid \boldsymbol{\Delta}_k^f \boldsymbol{X}' \leq \boldsymbol{\delta}_k^f\}$ for $k \in K^f$ with a certain matrix $\boldsymbol{\Delta}_k^f$ and a vector $\boldsymbol{\delta}_k^f$ with appropriate sizes. Furthermore, $\Psi_k^f$ and $\boldsymbol{\psi}_k^f$ for $k \in K^f$ are the $k$-th linear transformation matrix and the bias vector, respectively, and $K^f$ denotes the number of polytopes of a piecewise-linear function $f$.

Considering piecewise-assignment and piecewise-linear functions, the following properties straightforwardly hold:

• The concatenation of two or more piecewise-linear functions results in a piecewise-linear function.

• The composition of two or more piecewise-linear functions results in a piecewise-linear function.

• The composition of a piecewise-linear function and a piecewise-assignment function results in a piecewise-assignment function.

## 4.4 VAE-BASED AD AS PIECEWISE-ASSIGNMENT FUNCTION

In this subsection, we show that the VAE-based AD algorithm in Eq. (4) is a piecewise-assignment function by verifying that i) the reconstruction error in Eq. (1) is a piecewise-linear function, and ii) the thresholding in Eq. (2) is a piecewise-assignment function.

Most of basic operations and common activation functions used in the encoder and decoder networks can be represented as piecewise-linear functions in the form of Eq. (12). For example, the ReLU function is a piecewise-linear function. Operations like matrix-vector multiplication, convolution, and upsampling are linear, which categorizes them as special cases of piecewise-linear functions Furthermore, operations like max-pooling and mean-pooling can be represented in the form of Eq. (12). For instance, max-pooling of two variables can be expressed as $\max\{u, v\} = u \cdot I(u \geq v) + v \cdot I(v > u)$, which is a piecewise-linear function with $K^f = 2$. Consequently, the encoder and decoder networks of the VAE, composed or concatenated from piecewise-linear functions, form a piecewise-linear function. We note that this characteristic is not exclusive to our VAE; instead, it applies to the majority of CNN-type deep learning models[1].

Furthermore, the reconstruction error in Eq. (1) is also a piecewise-linear function. Specifically, let $f_{\text{abs}}$ be the absolute value function, which is clearly piecewise-linear function, $f_{\text{mm1}}$ be a function for multiplying the matrix $(I_n, -I_n)$ from the left, and $f_{\text{mm2}}$ be a function for multiplying the matrix $(I_n, I_n)^\top$ from the left. Then, the reconstruction error $E_i(\boldsymbol{X}) = |\boldsymbol{\mu_\theta}(\boldsymbol{\mu_\phi}(\boldsymbol{X})) - \boldsymbol{X}|_i$ is given as the $i^{\text{th}}$ element of the following compositions of multiple piecewise-linear functions:

$$f_{\text{abs}} \circ f_{\text{mm1}} \circ [\ \boldsymbol{\mu_\theta} \circ \boldsymbol{\mu_\phi} \quad I_n \ ] \circ f_{\text{mm2}}(\boldsymbol{X}).$$

The thresholding operation in Eq. (2) is clearly piecewise-assignment function. It means that the operation of detecting abnormal region $A_{\boldsymbol{X}}$ in Eq. (4) is composition of piecewise-linear function

---

[1]An example of components that do not exhibit piecewise linearity is nonlinear activation function such as the sigmoid function. However, since a one-dimensional nonlinear function can be approximated with high accuracy by a piecewise-linear function with sufficiently many segments, there are no practical problems.

and piecewise-assignment function, which results in a piecewise-assignment function. We summarize the aforementioned discussion into the following lemma.

**Lemma 1.** *The anomaly detection using VAE defined in Eq. equation 4, which uses piecewise-linear functions in the encoder and decoder network, is a piecewise-assignment function.*

Consequently, we can conduct the statistical test in equation 5 based on the selective $p$-value in equation 11 along with Theorem 1.

## 5 COMPUTATIONAL TRICKS

In this section, we demonstrate the procedure for efficiently computing the truncated intervals $\mathcal{Z}$ derived from Eq. equation 4. The identification of $\mathcal{Z}$ is challenging because the VAE-based AD is comprised of a substantial number of known piecewise-linear functions and a piecewise-assignment function. There are two difficulties: i) which indices of $k$ whose anomaly region is the same as the observed one, and ii) how to compute each truncated interval $[L_k^{\mathcal{A}}, U_k^{\mathcal{A}}]$. Our idea is to leverage parametric programming in conjunction with *auto-conditioning* to efficiently compute $\mathcal{Z}$. Specifically, we can identify only the necessary indices of $k$ and determining their respective intervals $[L_k^{\mathcal{A}}, U_k^{\mathcal{A}}]$. This enables us to bypass the unneeded computation of unnecessary components, thus saving computational time.

**Parametric Programming**   In the Theorem 1, the truncated intervals $\mathcal{Z}$ can be regarded as the intersections of the polytopes $\{P_k^{\mathcal{A}}\}_{k:A_k=A_{\boldsymbol{x}}}$ with the line $\boldsymbol{X} = \mathcal{Q}_{\boldsymbol{x}} + \Sigma\boldsymbol{\eta}(\boldsymbol{\eta}^\top\Sigma\boldsymbol{\eta})^{-1}Z$. This implies that determining the truncated intervals $\mathcal{Z}$ is accomplished by examining this specific line rather than the entire space. Alogorithm 1 outlines the procedure to identify $\mathcal{Z}$. The algorithm starts at $z_{\min}$ and search for the truncated intervals along the line until $z_{\max}$[2]. For each step, given $z$, the algorithm computes the lower bound $L_k^{\mathcal{A}}$ and upper bound $U_k^{\mathcal{A}}$ of the interval to which $z$ belongs to, as well as corresponding anomaly region $A_k = A_{\boldsymbol{X}(z)}$. The $L_k^{\mathcal{A}}$ and $U_k^{\mathcal{A}}$ are computed by the technique described in the next subsection. This procedure is commonly referred to as parametric programming known as parametric programming, which is a method to solve the optimization problem for parameters such as the lasso regularization path (Efron et al., 2004; Hastie et al., 2004; Karasuyama et al., 2012).

**Auto-Conditioning**   In line 4 of Algorithm 1, we utilize a technique referred to as *auto-conditioning*. Similar to *auto-differentiation*, this method leverages the fact that the entire computations of $L_k^{\mathcal{A}}$ and $U_k^{\mathcal{A}}$ executes a sequence of piece-wise linear operations. By applying the recursive rule repeatedly to these operations, $L_k^{\mathcal{A}}$ and $U_k^{\mathcal{A}}$ can be automatically computed. The details are deferred to Appendix A.3. This implies that by implementing the computational techniques for known piecewise-linear/assignment functions, we can automatically compute the truncation intervals and the anomaly region. This adaptability proves particularly advantageous when dealing with complex systems like Deep Neural Networks (DNNs), where frequent and detailed structural adjustments are often required. We note that the auto-conditioning technique is originally proposed in Miwa et al. (2023). However, the authors concentrate on a specific application of the saliency region, and no existing studies recognize its crucial application in VAE literature. In this paper, we prove that a VAE can be represented as a piecewise-assignment function, thus highlighting the crucial application of auto-conditioning in efficiently conducting the proposed VAE-AD Test.

## 6 EXPERIMENT

We demonstrate the performance of the proposed method. More details and results can be found in the Appendix A.5.

**Experimental Setup.** We compared the proposed method (VAE-AD Test) with OC (simple extension of SI literature to our setting), Bonferroni correction (Bonf) and naive method. More details can be found in Appendix A.5. We considered two covariance matrix structures:

---

[2]We set the $z_{\min} = -|T(\boldsymbol{x})| - 10\sigma$ and $z_{\max} = |T(\boldsymbol{x})| + 10\sigma$, where $\sigma$ is the standard deviation of test statistic. This is justified by the fact that the probability in the tails of the normal distribution can be considered negligible.

---

**Algorithm 1** Parametric Programming-Based SI

---

**Require:** $\boldsymbol{x}, z_{\min}, z_{\max}$
 1: Obtaine $A_{\boldsymbol{x}}$ and compute $\boldsymbol{\eta}$.
 2: $z \leftarrow z_{\min}$ and $\mathcal{Z} \leftarrow \emptyset$
 3: **while** $z \leq z_{\max}$ **do**
 4:    Compute $L_k^{\mathcal{A}}, U_k^{\mathcal{A}}$, and $A_k$ respect to $z$ by *auto-conditioning* (see Appendix A.3).
 5:    **if** $A_k = A_{\boldsymbol{x}}$ **then**
 6:       $\mathcal{Z} \leftarrow \mathcal{Z} \cup [L_k^{\mathcal{A}}, U_k^{\mathcal{A}}]$
 7:    **end if**
 8:    $z \leftarrow U_k^{\mathcal{A}} + \delta$, where $\delta$ is a small positive number.
 9: **end while**
10: $p_{\text{selective}} \leftarrow$ equation 11 with Threorem 1
**output** $p_{\text{selective}}$ and $A_{\boldsymbol{x}}$

---

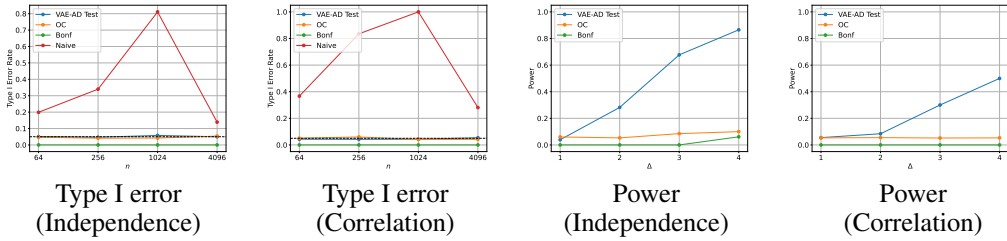

| Type I error (Independence) | Type I error (Correlation) | Power (Independence) | Power (Correlation) |

Figure 2: Type I errors (false positive detection rates) and powers (true positive detection rates) of the proposed VAE-AD Test and three baselines, Naive, OC and Bonf in Indepence and Correlation setting. Naive test, which does not consider the fact that abnormal regions are selected in a data-driven manner, fails to control the Type I error, failing to meet the requirements of a statistical test. On the other hand, the proposed method, VAE-AD Test, and two other baselines, OC and Bonf, all successfully control the Type I error at 0.05 in all settings. The power of the proposed VAE-AD Test is significantly larger than two baselines, OC and Bonf in all problem settings.

- $\Sigma = I_n$ (Independence)

- $\Sigma = \text{AR}(1) \otimes \text{AR}(1)$ (Correlation) where $\text{AR}(1)$ is the first-order autoregressive matrix $\{\text{AR}(1)\}_{ij} \in \mathbb{R}^{\sqrt{n} \times \sqrt{n}} = 0.25^{|i-j|}$ and $\otimes$ is kronecker dot.

To examine the type I error rate, we generated 1000 null images $\boldsymbol{X} = (X_1, \ldots, X_n)$, where $\boldsymbol{s} = \boldsymbol{0}$ and $\boldsymbol{\epsilon} \sim N(\boldsymbol{0}, \Sigma)$, for each $n \in \{64, 256, 1024, 4096\}$. To examine the power, we set $n = 256$ and generated 1000 images in which $\boldsymbol{\epsilon} \sim N(\boldsymbol{0}, \Sigma)$, the signals $s_i = \Delta$ for any $i \notin \mathcal{S}$ where $\mathcal{S}$ is the "true" anomaly region whose location is randomly determined, and $s_i = 0$ for any $i \notin \mathcal{S}$. We set $\Delta \in \{1, 2, 3, 4\}$. In all experiments, we set the threshold $\lambda = 1.2$ for the anomaly detection, and the significance level $\alpha = 0.05$. We also apply mean filtering to the reconstruction error to enhance the anomaly detection performance.

**Numerical results.** The results of type I error rate and power are shown in Fig. 2. The VAE-AD Test, OC, and Bonf successfully controlled the type I error rate in the both cases of independence and correlation, whereas the naive method could not. Since the naive method failed to control the type I error, we no longer considered its power. The power of the VAE-AD Test was the highest among the methods that controlled the type I error. The Bonferroni method has the lowest power because it is conservative due to considering the huge number of all possible hypotheses. OC also has low power because it considers extra conditioning, which causes the loss of power.

**Real data experiments.** We examined the brain image dataset extracted from Buda et al. (2019), which includes 939 and 941 images with and without tumors, respectively. The results of statistical testing for images without tumor and with tumor are presented in Figs. 3 and 4. The naive $p$-value is small even in cases where no tumor region exists in the image. This indicates that the naive $p$-value

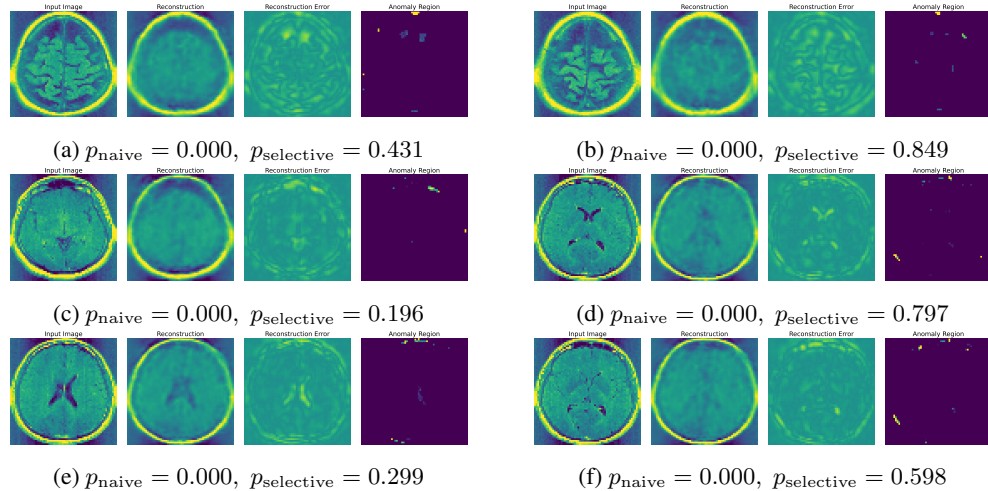

Figure 3: Anomaly detection for images without tumor. The naive $p$-values are 0.000 in all settings, incorrectly detecting abnormalities. However, the selective $p$-values based on the proposed VAE-AD Test are all large enough, correctly identifying the absence of abnormalities.

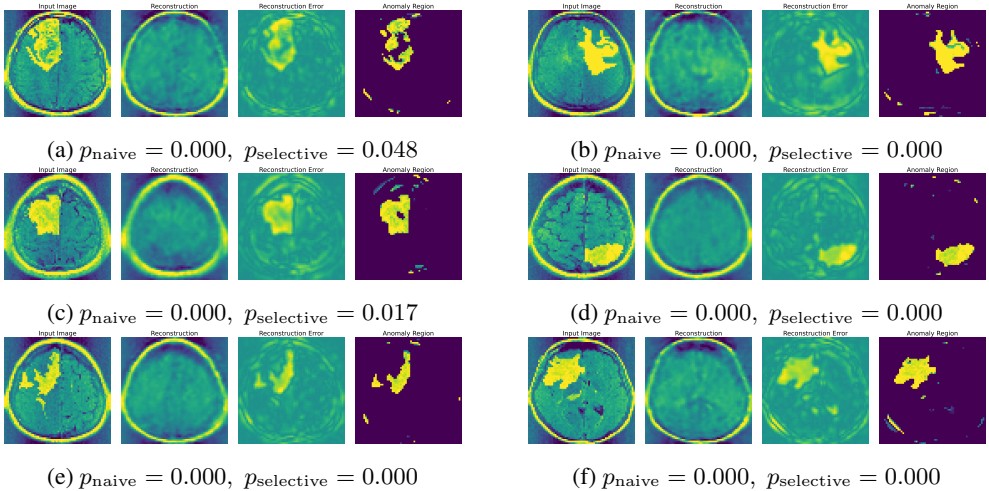

Figure 4: Anormaly detection for images with tumor. In all settings, both the naive $p$-values and the selective $p$-values are low, correctly identifying the abnormalities (although naive $p$-values are invalid statistical tests because it fails to control type I errors).

cannot be used to quantify the reliability of the result of anomaly detection using VAE. With the proposed selective $p$-values, we successfully identified false and true positive detections.

## 7 CONCLUSIONS, LIMITAIONS AND FUTURE WORKS

We introduced a novel statistical testing framework for AD task using deep learning model. We developed a valid statistical test for VAE-based AD using CSI. We believe that this study stands as a significant step toward reliability of deep learning model-based decision making. There are several constraints on the class of problems where CSI can be applied, so new challenges arise when applying VAEs to other types of neural networks. Additionally, we selected simple options for defining the anomalous region and the test statistic, but it is unknown whether the same framework can be applied to more complex options. Furthermore, as the size of the VAE network increases, the computational cost of calculating the selective $p$-value also increases, necessitating the development of cost reduction methodologies such as parallelization.

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

## A  APPENDIX

### A.1  THE DETAILS OF VAE

We used the architecture of the VAE as shown in Figure 5 and set $m = 10$ as a dimensionality of the latent space. We used ReLU as an activation function for the encoder and decoder. We generated 1000 images from $N(\mathbf{0}, I_n)$ as normal images and trained the VAE with these images, and used Adam Kingma & Ba (2017) as an optimizer.

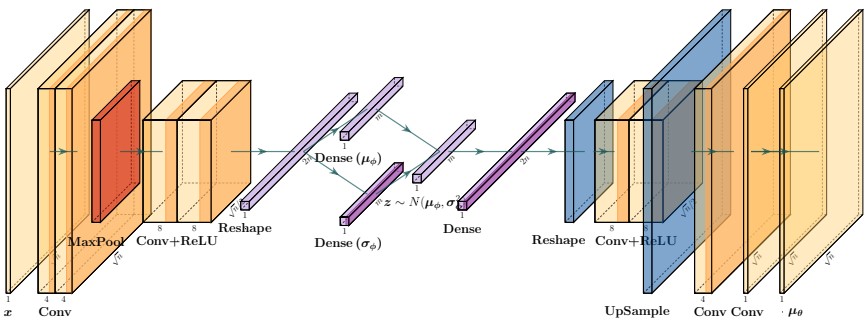

Figure 5: Architecture of the VAE.

### A.2  PROOF OF THEOREM 1

*Proof.* The theorem is based on the Lemma 3.1 in Chen & Bien (2020). By the definition of the piecewise-assignment function, the conditional part, $\{M_{\boldsymbol{X}} = M_{\boldsymbol{x}}\}$ can be characterized as the union of polytopes,

$$\{M_{\boldsymbol{X}} = M_{\boldsymbol{x}}\} = \bigcup_{k:M_k = M_{\boldsymbol{x}}} P_k^{\mathcal{M}}.$$

By substituting $\boldsymbol{X}(Z) = \mathcal{Q}_{\boldsymbol{x}} + \Sigma\boldsymbol{\eta}(\boldsymbol{\eta}^{\top}\Sigma\boldsymbol{\eta})^{-1}Z$ into the polytopes $P_k^M$, we obtain the truncated intervals $\mathcal{Z}$ in the lemma. For the set $k$ such that $M_k = M_{\boldsymbol{x}}$, we have $\mathcal{Q}_{\boldsymbol{X}} \perp Z$ by orthogonality of $\mathcal{Q}_{\boldsymbol{X}}$ and $\boldsymbol{\eta}$ and by the properties of the normal distribution. Hence, we obtain

$$Z \mid \{M_{\boldsymbol{X}} = M_{\boldsymbol{x}}, \mathcal{Q}_{\boldsymbol{X}} = \mathcal{Q}_{\boldsymbol{x}}\} \stackrel{d}{=} Z \mid \{Z \in \mathcal{Z}, \mathcal{Q}_{\boldsymbol{X}} = \mathcal{Q}_{\boldsymbol{x}}\}$$
$$\stackrel{d}{=} Z \mid \{Z \in \mathcal{Z}\} \; (\because \mathcal{Q}_{\boldsymbol{X}} \perp Z)$$

There is no randomness in $\mathcal{Z}$,

$$Z \mid \{M_{\boldsymbol{X}} = M_{\boldsymbol{x}}, \mathcal{Q}_{\boldsymbol{X}} = \mathcal{Q}_{\boldsymbol{x}}\} \sim TN(\boldsymbol{\eta}^{\top}\boldsymbol{\mu}, \boldsymbol{\eta}^{\top}\Sigma\boldsymbol{\eta}; \mathcal{Z}).$$

□

### A.3 THE DETAILS OF AUTO-CONDITIONING

This section demonstrates the auto-conditioning algorithm, utilized to compute the truncated intervals $[L_k^{\mathcal{A}}, U_k^{\mathcal{A}}]$ and the corresponding anomaly region $A_k$ respect to the $z$ in Algorithm 1. The algorithm is introduced for the piecewise-assignment function, which is composed of piecewise-linear functions and a piecewise-assignment function.

It is conceptualized as a directed acyclic graph (DAG) that delineates the processing of input data, similar to a computational graph in auto-differentiation. In this graph, the nodes symbolize the piecewise-linear and piecewise-assignment functions, each with an input and output edge to represent the function compositions. It should be noted that the node such as $\boldsymbol{\mu_\theta}$ and $\boldsymbol{\mu_\theta}$, may replace the other DAG express the piecewise-linear/assignment function of the node since it can be represented as the composition and concatenation of array of simpler piecewise-linear/assignment functions. The level of simplicity for a function of a node can be determined based on what is most convenient for the implementation. A special node, representing the concatenation of two piecewise-linear functions, features two input edges and one output edge. Figure 6 shows the directed acyclic graph of the anomaly detection using VAE in Eq. equation 4.

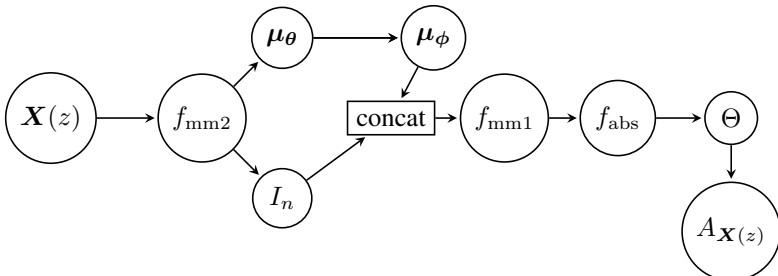

Figure 6: The directed acyclic graph of the anomaly detection using VAE $\mathcal{A} : \mathbb{R}^n \to 2^{|n|}$ defined in Eq. equation 4. Circles represent the piecewise-linear functions and the piecewise-assignment function. The rectangle represents the concatenation of piecewise-linear functions. The edges represent the composition of piecewise-linear functions.

#### A.3.1 UPDATE RULES FOR THE NODES OF THE PIECEWISE-ASSIGNMENT FUNCTIONS

The computation of the interval $[L_k^{\mathcal{A}}, L_k^{\mathcal{A}}]$ is defined in a recursive way. The output of the node $f : \mathbb{R}^l \to \mathbb{R}^m$ in the DAG are denoted as $\boldsymbol{a}_f, \boldsymbol{b}_f \in \mathbb{R}^m$ and $L_f, U_f \in \mathbb{R}$.

**Update rule for the initial node.** At first, the output of the initial node $\boldsymbol{X}(z)$ of the directional graph denoted as $f_0$ for notational convention, are defined as $\boldsymbol{a}_{f_0} = \mathcal{Q}_{\boldsymbol{x}}$, $\boldsymbol{b}_{f_0} = \Sigma\boldsymbol{\eta}(\boldsymbol{\eta}^\top\Sigma\boldsymbol{\eta})^{-1}$, $L_{f_0} = -\infty$, and $U_{f_0} = \infty$. It should be noted here that $\boldsymbol{X}(z) = \boldsymbol{a}_{f_0} + \boldsymbol{b}_{f_0} z$ is the line appeared in the proof of Theorem 1 in Section A.2.

**Update rule for the node of the piecewise-linear functions.** Let us consider the output for the node $g$ whose input is the output of the node $f$ in the DAG. The inputs of the $g$'s node (i.e. output of node $f$) are denoted as $\boldsymbol{a}_f, \boldsymbol{b}_f, L_f$ and $U_f$. $\boldsymbol{a}_f$ is the summed point vector added in the piecewise-linear functions until reaching $f$, $\boldsymbol{b}_f$ is the direction vector corresponding to $z$, multiplied in the piecewise-linear functions until reaching to $f$. Then, the output of the piecewise-linear function $f$ is represented as $\boldsymbol{a}_f + \boldsymbol{b}_f z$. $L_f$ and $U_f$ are the lower and upper bounds of the interval obtained at the piecewise-linear function $f$. The output of the node $g$ is defined as follows: 1) Check the index $j$ such that the output of $f$ within the polytope of: $P_j^g \ni \boldsymbol{a}_f + \boldsymbol{b}_f z$. 2) Compute the point vector $\boldsymbol{a}_g$ and the direction vector $\boldsymbol{b}_g$ of the piecewise-linear function $g$ with the index $j$,

$$\boldsymbol{a}_g = \boldsymbol{\Psi}_j^g \boldsymbol{a}_f + \boldsymbol{\psi}_j^g, \ \boldsymbol{b}_g = \boldsymbol{\Psi}_j^g \boldsymbol{b}_f. \tag{13}$$

3) Compute the lower and upper bounds of the interval $L_g$ and $U_g$ with the index $j$,

$$L = \max_{k:(\boldsymbol{\beta}_j^g)_k > 0} \frac{(\boldsymbol{\alpha}_j^g)_k}{(\boldsymbol{\beta}_j^g)_k}, \ U = \min_{k:(\boldsymbol{\beta}_j^g)_k < 0} \frac{(\boldsymbol{\alpha}_j^g)_k}{(\boldsymbol{\beta}_j^g)_k},$$

where $\boldsymbol{\alpha}_j^g = \boldsymbol{\delta}_j^g - \boldsymbol{\Delta}_j^g \boldsymbol{a}_f$ and $\boldsymbol{\beta}_j^f = \boldsymbol{\Delta}_j^g \boldsymbol{b}_f$. 4) Take the intersection of the interval $[L_f, U_f] \cap [L, U]$ as the interval $[L_g, U_g]$ of the piecewise-assignment function $g$ as

$$L_g = \max(L_f, L), \ U_g = \min(U_f, U).$$

This update rule is obtained from the Lemma 2 in Miwa et al. (2023).

**Update rule for the nodes of concatenation of two piecewise-linear functions.** Let us consider the concatenation node of two piecewise-linear functions $f$ and $g$ denoted as concat. Let the inputs of the node be $\boldsymbol{a}_f, \boldsymbol{b}_f, L_f$ and $U_f$ from the node $f$ and $\boldsymbol{a}_g, \boldsymbol{b}_g, L_g$ and $U_g$ from the node $g$. The output of the concatenation node, $\boldsymbol{a}_{\text{concat}}, \boldsymbol{b}_{\text{concat}}, L_{\text{concat}}$ and $U_{\text{concat}}$ are defined as follows: 1) Concatenate the vector outputs of nodes $f$ and $g$

$$\boldsymbol{a}_{\text{concat}} = \begin{bmatrix} \boldsymbol{a}_f \\ \boldsymbol{a}_g \end{bmatrix}, \ \boldsymbol{b}_{\text{concat}} = \begin{bmatrix} \boldsymbol{b}_f \\ \boldsymbol{b}_g \end{bmatrix}.$$

2) Take intersection of the interval $[L_f, U_f] \cap [L_g, U_g]$ as

$$L_{\text{concat}} = \max(L_f, L_g), \ U_{\text{concat}} = \min(U_f, U_g).$$

**Update rule for the final node.** At the final node $\Theta$ which is the piecewise-assignment function, it takes the same input as the node of piecewise-linear functions and outputs are the same except for the $\boldsymbol{a}_\Theta$ and $\boldsymbol{b}_\Theta$. 1) It computes the index $j$ such that the input falls into the polypotopes of $P_j^\Theta$. 2) Then, the anomaly region $A_j$ is obtained instead of Eq. equation 13 in the update rule for the node of piecewise-linear functions. 3) The computation of lower bounds $L_\Theta$ and the upper bounds $U_\Theta$ are the same as the update rule for the node of piecewise-linear functions. The output of the final node are the anomaly region $A_j$, the lower bounds $L_\Theta$ and the upper bounds $U_\Theta$.

Then, apply the above update rule to the directional graph of the piecewise-assignment function from the initial node $f_0$ to the final node $\Theta$. Consequently, the auto-conditioning algorithm computes the lower and upper bounds of the interval as the outputs of final node $L_k^{\mathcal{A}} = L_\Theta, L_k^{\mathcal{A}} = L_\Theta$ and $A_k = A_j$.

### A.4 IMPLEMENTATION

We implemented the auto-conditioning algorithm described above in Python using the tensorflow library. The codes construct the DAG of the piecewise-assignment function automatically from the trained Keras/tensorflow model. Then, we do not need further implementation to conduct CSI for each specific DNN model. This indicates that even if we change the architecture or adjust the hyper-parameters and retrain the DNN models, we can conduct the CSI without additional implementation.

### A.5 EXPERIMENTAL DETAILS

**Methods for comparison.** We compared our proposed method with the following methods:

- VAE-AD Test: our proposed method.

- OC: our proposed method conditioning on the only one polytope to which the observed image belongs $\boldsymbol{x} \in P_k^{\mathcal{A}}$. This method is computationally efficient; however, its power is low due to over-conditioning.

- Bonf: the number of all possible hypotheses are considered to account for the selection bias. The $p$-value is computed by $p_{\text{bonf}} = \min(1, p_{naive} \times 2^n)$

- Naive: the conventional method is used to compute the $p$-value.

**Experiment for robustness.** We evaluate the robustness of our proposed methodology in terms of Type I error control, specifically under conditions where the noise distribution deviates from the Gaussian assumption. We investigate this robustness by applying our method across a range of non-Gaussian noise distributions, including:

- Skew normal distribution (*skewnorm*)

- Exponential normal distribution (*exponorm*)

- Generalized normal distribution with steep tails (*gennormsteep*)

- Generalized normal distribution with flat tails (*gennomflat*)

- Student's $t$ distribution ($t$)

We commence our analysis by identifying noise distributions from the aforementioned list that have a 1-Wasserstein distance of $\{0.01, 0.02, 0.03, 0.04\}$ relative to the standard normal distribution $N(0, 1)$. Subsequently, we standardize these noise distributions to ensure a mean of 0 and a variance of 1. Setting the sample size to $n = 256$, we generate 1000 samples from the selected distributions and apply hypothesis testing to each sample to obtain the Type I error rate. This process is conducted at significance levels $\alpha = \{0.05, 0.10\}$. The results are shown in Fig. 7. Our method still maintains good performance in type I error rate control.

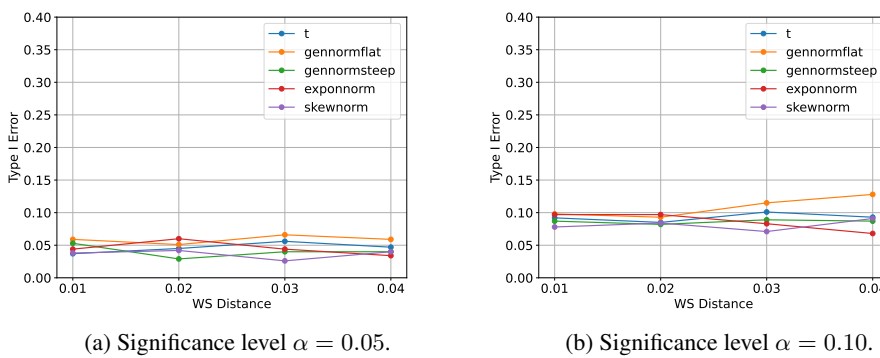

(a) Significance level $\alpha = 0.05$.      (b) Significance level $\alpha = 0.10$.

Figure 7: Robustness of type I error control.

**More results on brain image dataset.** Additional results are shown in Figs. 8 and 9.

**Computational resources used in the experiments.** All numerical experiments were conducted on a computer with a 56-core 2.00GHz CPU, eight RTX-A6000 GPUs, and 1024GB of memory.

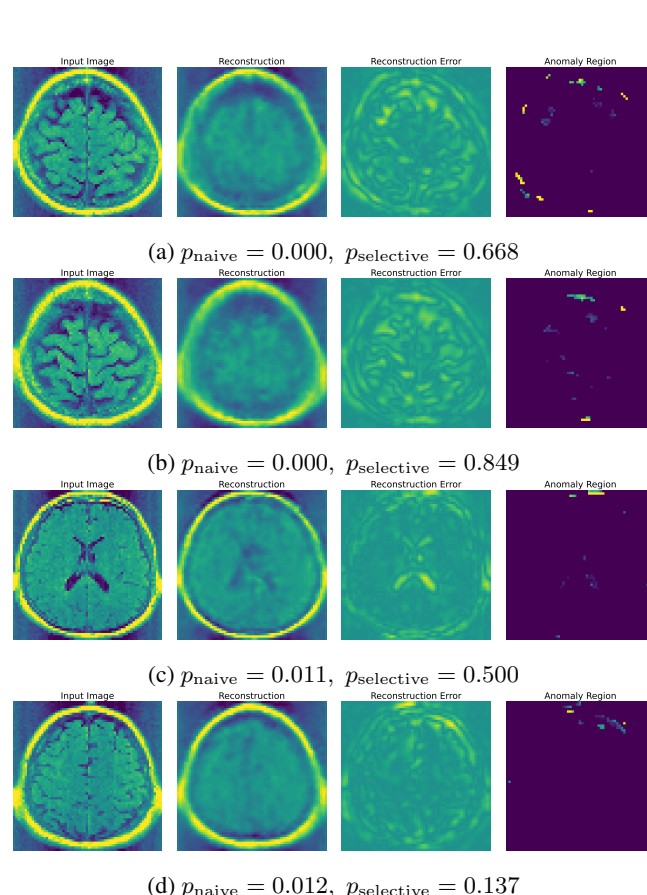

(a) $p_{\mathrm{naive}} = 0.000, \ p_{\mathrm{selective}} = 0.668$

(b) $p_{\mathrm{naive}} = 0.000, \ p_{\mathrm{selective}} = 0.849$

(c) $p_{\mathrm{naive}} = 0.011, \ p_{\mathrm{selective}} = 0.500$

(d) $p_{\mathrm{naive}} = 0.012, \ p_{\mathrm{selective}} = 0.137$

Figure 8: Anomaly detection for image without tumor.

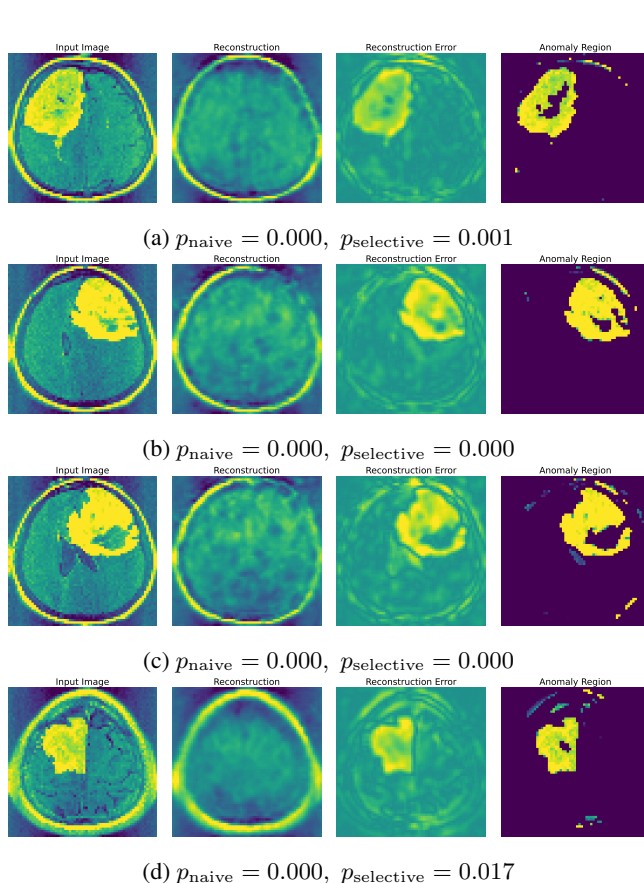

(a) $p_{\text{naive}} = 0.000$, $p_{\text{selective}} = 0.001$

(b) $p_{\text{naive}} = 0.000$, $p_{\text{selective}} = 0.000$

(c) $p_{\text{naive}} = 0.000$, $p_{\text{selective}} = 0.000$

(d) $p_{\text{naive}} = 0.000$, $p_{\text{selective}} = 0.017$

Figure 9: Anormaly detection for image with tumor.

