# OpenReview forum: "Statistical Test for Anomaly Detections using Variational Auto-Encoders by Selective Inference"
_ICLR.cc/2025/Conference — ICLR 2025 Conference Withdrawn Submission_

### Official Review · Reviewer_AtnE · 2024-10-24

**Soundness:** 3
**Presentation:** 2
**Contribution:** 2
**Rating:** 5
**Confidence:** 3

**Summary:**

This paper proposes the VAE-AD Test, the novel appraoch for statistically evaluating the reliability of anomaly detection results using Variational Autoencoders (VAE). The proposed method introduces a test statistic based on the reconstruction error of the VAE, and by applying selective inference to assign appropriate p-values to the detected anomalies, it can theoretically control the false detection rate. The effectiveness of the proposed method is demonstrated, particularly in experiments using medical images.

**Strengths:**

The proposed method provides theoretical guarantees for the reliability of anomaly detection using VAE, leveraging techniques that are well-suited to the characteristics of VAE, such as Piecewise-Assignment Functions and Piecewise-Linear Functions. Additionally, it introduces interesting techniques for efficiently computing the proposed method. The proposed method makes a significant contribution to the reliability of detection results and is highly useful in critical decision-making tasks, such as in the medical field.

**Weaknesses:**

The proposed method is interesting, but there are some unclear points. I will write the details in the Questions section, so please refer to it.

**Questions:**

1. Why does the proposed method use the VAE? Since it is based on the reconstruction error, I think an Autoencoder would be more suitable. Since the objective function of VAE, ELBO, includes a KL divergence term in addition to the reconstruction error, the reconstruction quality may not be very good, as seen in Figure 1.
2. Conversely, what is the reason for using reconstruction error? With the VAE, it is possible to calculate probability values using importance sampling. I believe probability values would be more appropriate as anomaly scores than reconstruction error. ELBO could also be used as an alternative to probability values.
3. As in Eq. (3), Gaussian noise is being added to the original data. As mentioned at the beginning of Section 6, its covariance matrix is set in two different ways. It seems that the noise based on the identity matrix shows better results, but noise following the identity matrix would be large if the image is normalized, and small if it is not. How would the performance change if the variance were scaled by a constant, such as $\beta I$?
4. In Eqs. (5) and (7), the difference in the mean values of each pixel between the normal and anomaly regions is used as the test statistic. I don't fully understand the reason for adopting this test statistic, so could you explain it? It seems obvious that the pixel values would differ between the normal and anomaly regions.

---

> ### Author Response · Authors · 2024-11-23
>
> We thank the reviewer for your feedback.
>
> > 1. Why does the proposed method use the VAE? Since it is based on the reconstruction error, I think an Autoencoder would be more suitable. Since the objective function of VAE, ELBO, includes a KL divergence term in addition to the reconstruction error, the reconstruction quality may not be very good, as seen in Figure 1.
>
> Our objective is to test the anomalous regions detected by VAE, rather than to perfectly reconstruct the input image. As shown in Figure 1, the VAE-based anomaly detection (VAE-AD) effectively identifies these regions. While comparing autoencoders (AEs) and VAEs for reconstruction-based anomaly detection is not the primary focus of our paper, we believe that VAEs are better suited for this task due to the regularization provided by the KL-divergence term in the Evidence Lower Bound (ELBO). This regularization promotes a smoother and more robust latent space for normal images, allowing images with anomalous regions to be reconstructed as normal, which proves effective for detecting anomalous regions in the image.
>
> > 2. Conversely, what is the reason for using reconstruction error? With the VAE, it is possible to calculate probability values using importance sampling. I believe probability values would be more appropriate as anomaly scores than reconstruction error. ELBO could also be used as an alternative to probability values.
>
> We agree that the use of probability values as an alternative approach could offer potential advantages. However, based on prior studies related to VAE-based anomaly detection for brain tumor detection [1], reconstruction error has been shown to achieve competitive performance compared to methods using the gradient of ELBO or KL-divergence. Given its simplicity and practicality, we have chosen reconstruction error as the starting point for our approach in this study. While we acknowledge the potential advantages of probability-based methods and ELBO, exploring these alternatives represents a promising direction for future work, and we plan to investigate them in subsequent studies.
>
> [1] Zimmerer, D., Isensee, F., Petersen, J., Kohl, S., & Maier-Hein, K. (2019). Unsupervised anomaly localization using variational auto-encoders. In Medical Image Computing and Computer Assisted Intervention–MICCAI 2019: 22nd International Conference, Shenzhen, China, October 13–17, 2019, Proceedings, Part IV 22 (pp. 289-297). Springer International Publishing.
>
> > 3. As in Eq. (3), Gaussian noise is being added to the original data. As mentioned at the beginning of Section 6, its covariance matrix is set in two different ways. It seems that the noise based on the identity matrix shows better results, but noise following the identity matrix would be large if the image is normalized, and small if it is not. How would the performance change if the variance were scaled by a constant, such as $\beta \mathbf{I}$
>
> We would like to clarify the assumption that the original data(image) is a random vector containing a true signal, $\mathbf{s}$ and observed with Gaussian noise, $\mathbf{\epsilon}$ rather than the noise being added to the original data.
> Therefore, the normalization do not affect the scale of variance, as normalizing the image also scales its variance.
> We hypothesize that the increasing the variance deteriorates the power, following an analogy with the two-sample test in statistics, although the Type I Error is controlled by significant level $\alpha$.
>
>
> > 4. In Eqs. (5) and (7), the difference in the mean values of each pixel between the normal and anomaly regions is used as the test statistic. I don't fully understand the reason for adopting this test statistic, so could you explain it? It seems obvious that the pixel values would differ between the normal and anomaly regions.
>
> We would like to point out the normal region $A_{X}$ and the abnormal region $A^c_{X}$ are the regions diagnosed by VAE-based-AD, not true ones.
> As we mentioned above, the data is random vector, containing true signals, $\mathbf{s}$ observed with Gaussian noise, then, the true normal region might be incorrectly detected as a abnormal regions by VAE-based-AD.
> It is natural to assume that the average true signal value of each pixel in the true normal region is consistent but varies in the true abnormal region. Therefore, we defined the null hypothesis, $H_0$, as the scenario where the true normal region is detected as the abnormal region, $A$, and the alternative hypothesis, $H_1$, as the scenario where the true abnormal region is detected as the abnormal region, $A$, by VAE-based-AD, as stated in Eq. (5).
> These hypotheses can be tested using the test statistic defined in Eq. (7).

---

> > ### Comment · Reviewer_AtnE · 2024-11-26
> >
> > Thank you for your reply.
> > I have a better understanding of questions 3 and 4.
> >
> > On the other hand, I think there is not enough discussion about the point of using only the reconstruction error of VAE.
> > For example, the following papers investigate the performance of anomaly detection for VAE and autoencoders.
> > I think it is necessary to discuss this in the related work section after clarifying the relevance to these papers.
> >
> > I will maintain my score.
> >
> > [1] Nalisnick, Eric, et al. "Do deep generative models know what they don't know?." arXiv preprint arXiv:1810.09136 (2018).
> >
> > [2] Havtorn, Jakob D., et al. "Hierarchical vaes know what they don’t know." International Conference on Machine Learning. PMLR, 2021.
> >
> > [3] Choi, Hyunsun, Eric Jang, and Alexander A. Alemi. "Waic, but why? generative ensembles for robust anomaly detection." arXiv preprint arXiv:1810.01392 (2018).
> >
> > [4] Yoon, Sangwoong, Yung-Kyun Noh, and Frank Park. "Autoencoding under normalization constraints." International Conference on Machine Learning. PMLR, 2021.

---

### Official Review · Reviewer_Mr87 · 2024-11-04

**Soundness:** 2
**Presentation:** 2
**Contribution:** 3
**Rating:** 5
**Confidence:** 4

**Summary:**

This paper proposes the VAE-AD Test, a statistical framework designed to assess the validity of detected anomalies in pixel reconstruction errors using a Variational Autoencoder (VAE). The authors argue that a significant limitation in the literature on VAE-based anomaly detection is the lack of rigorous statistical measures to validate detected anomaly regions, which is critical for high-stakes applications like medical diagnostics. To address this gap, the authors formulate a statistical test based on differences in average reconstruction errors inside versus outside detected anomalous regions, leveraging Conditional Selective Inference (CSI) to compute valid post-selection p-values via Piecewise-Assignment Functions.

**Strengths:**

The paper addresses an important problem in the field of deep learning-based anomaly detection by proposing a method for validating detected anomalies using the same dataset. Applying Conditional Selective Inference to assess the reliability of anomaly detection with VAEs fills a notable gap in this area, providing a statistical approach to validate model outputs in scenarios where reliable decisions are essential.

**Weaknesses:**

•  Defining Anomalous Regions: The method defines an anomalous region as the set of pixels with reconstruction errors exceeding a user-defined threshold. This reliance on a threshold could reduce the method's usability in practice, as users may not know the appropriate threshold value. Typically, anomaly detection methods focus first on detecting or identifying anomalous regions (i.e., selecting λ, while a secondary goal is evaluating whether there is sufficient evidence to confirm those regions as true positives.

•  Region Constraints: The authors’ approach allows for detection of any subset of pixels (i.e., it considers all elements in the power set). Many region-based anomaly detection methods impose constraints to ensure detected anomalies are meaningful. For instance, in the case of brain tumor images, there is no restriction to prevent the selection of a scattered set of pixels that do not form a contiguous region, which may occur with a large enough λ . Conversely, a small λ could lead to selecting all pixels except for a few, producing results that are challenging to interpret. This underlines the importance of selecting λ judiciously to achieve meaningful region detection as a preliminary step.

•  Statistical Test Design: The authors’ null hypothesis H0 posits that the average reconstruction error inside the anomalous region is the same as outside. However, their test statistic actually tests whether the average reconstruction error within the region differs from a theoretical expectation under their parametric model (\eta^T*X, equation 7). This subtle difference implies that even if the in-region and out-region errors are not significantly different, the test could reject H0 if the parametric model's expected reconstruction error deviates from the observed value. Similarly, it could fail to reject H0 even when in-region and out-region errors differ, if these deviations do not align with the model’s theoretical expectations.

•  Section 4 Readability: Section 4, which appears to be the critical contribution, is dense and challenging to follow. The writing assumes familiarity with advanced concepts, placing a high cognitive load on readers who wish to fully understand the authors' approach. A clearer breakdown of the key steps, possibly with illustrative examples, could make this section more accessible and persuasive.

•  Related Work: This work has a similar objective to the literature on scan statistics, which aims to detect and test for “anomalous” regions. For example [1,2] tests for difference in vs out the region (which is the authors' original hypothesis setup), [3] extends this to observed vs expected (which is more consistent with the authors’ test statistic) and focuses efficiently finding the most anomalous subset of data points (which would equate to selecting the correct value of lambda). [4] Seems to extend some of the ideas of [3] to medical images as well. And if we alternatively consider the image to be an adjacency graph of pixels you’d have  [5,6,7]. To be clear, none of these use VAE, but  I also think for the authors VAE just represents a vehicle to measure deviations, upon which they then impose parametric assumptions. To be clear, I think VAE is a good vehicle to capture deviation, but given the use of VAE is not new to this work, the key contribution is the detection/testing of the anomalies produced by a VAE, so I am comparing the authors chosen approach to the scan approaches that making the same parametric assumptions could be applied to the deviations of the VAE

•  Experimental Limitations: The experiments provide limited insights into the practical benefits and drawbacks of this approach. Since the authors make no theoretical claims about properties other than the validity of the selective p-value, and given that the motivation is high-stakes AD, the experiments are crucial. However, they currently offer minimal information. For instance, it’s unclear how the threshold λ=1.2 was selected, which could offer practical guidance. Additionally, exploring the effects of different λ values on detection outcomes would provide valuable context: what happens when it is too small or too large. The graphs are somewhat difficult to interpret, lack confidence intervals, and, beyond Type I error and power, do not demonstrate the accuracy of detected regions compared to true anomaly regions.


References
[1] Kulldorff, M. (1997). A spatial scan statistic. Communications in Statistics - Theory and Methods, 26(6), 1481–1496. https://doi.org/10.1080/03610929708831995

[2] Kulldorff, M., Huang, L., Pickle, L., & Duczmal, L. (2006). An elliptic spatial scan statistic. Statistics in Medicine, 25, 3929–3943.

[3] Neill, D. B. (2012). Fast Subset Scan for Spatial Pattern Detection. Journal of the Royal Statistical Society: Series B (Statistical Methodology), 74(2), 337–360. https://doi.org/10.1111/j.1467-9868.2011.01014.x

[4] Somanchi, S., Neill, D. B., & Parwani, A. V. (2018). Discovering anomalous patterns in large digital pathology images. Statistics in Medicine, 37(25), 3599–3615.

[5] Patil, G. P., & Taillie, C. (2004). Upper Level Set Scan Statistic for Detecting Arbitrarily Shaped Hotspots. Environmental and Ecological Statistics, 11(3), 183–197.

[6] Speakman, S., McFowland, E., & Neill, D. B. (2015). Scalable Detection of Anomalous Patterns with Connectivity Constraints. Journal of Computational and Graphical Statistics, 24(4), 1014–1033.

[7] Tango, T., & Takahashi, K. (2005). A Flexibly Shaped Spatial Scan Statistic for Detecting Clusters. International Journal of Health Geographics, 4, 11.

**Questions:**

•  Applicability of CSI with Alternative Region Selection: If a more sophisticated region detection approach (like those in scan statistics) were used to detect anomalous regions based on reconstruction errors, could the authors’ CSI method still be applied? In other words, does CSI impose constraints on the region selection process, or could it flexibly accommodate other detection methods?

•  Comparison to Randomization Testing: Many anomaly detection methods assess validity through randomization testing (which could be performed here, given the authors’ parametric assumptions under H0) by generating samples of data under H0, computing the test statistic in these null data samples, and comparing the test statistic in the original data to those from the null data to compute its empirical p-value. Conceptually and theoretically what additional benefits does CSI provide over this approach? Would it be possible to compare CSI with randomization testing as was done with the naive p-value?

---

> ### Author Response · Authors · 2024-11-23
>
> > Defining Anomalous Regions: The method defines an anomalous region as the set of pixels with reconstruction errors exceeding a user-defined threshold.
>
> > Region Constraints: The authors’ approach allows for detection of any subset of pixels (i.e., it considers all elements in the power set).
>
> Our proposed method theoretically guarantees control over the Type I error rate at any user-specified significance level $\alpha$ irrespective of the specific value of $\lambda$ or the method used to determine it. Consequently, practitioners can use commonly adopted approaches for setting $\lambda$, such as optimizing it based on a validation dataset. While the choice and determination of $\lambda$ are important considerations, a comprehensive investigation of this topic lies beyond the primary scope of this study.
>
> > Statistical Test Design: The authors’ null hypothesis H0 posits that the average reconstruction error inside the anomalous region is the same as outside.
>
> We believe there may be some misunderstanding. To clarify, our null hypothesis $H_0$ focuses on the average signal values of the pixels in the *original image*, not the reconstruction error. Specifically, it states that the average signal values do not differ between the anomaly region and the area outside the anomaly region. Furthermore, we believe the test statistic defined in Eq. (7) is a straightforward derivation from the hypothesis.  The null hypothesis $H_0$ in Eq. (5) is expressed as $\mathbf{\eta}^\top \mathbf{s} = 0$, where $\mathbf{\eta}$ is same as in Eq. (7).
> This leads directly to the test statistic by substituting the true signal $\mathbf{s}$ with its observed value $\mathbf{X}$. Thus, the test statistic is fully consistent with our hypothesis.
>
> > Related Work: This work has a similar objective to the literature on scan statistics, which aims to detect and test for “anomalous” regions. For example [1,2] tests for difference in vs out the region (which is the authors' original hypothesis setup), [3] extends this to observed vs expected (which is more consistent with the authors’ test statistic) and focuses eficiently finding the most anomalous subset of data points (which would equate to selecting the correct value of lambda).
>
> We thank the reviewer for highlighting the connection to scan statistics and for providing relevant references. While we acknowledge the shared objective of detecting and testing anomalous regions, our focus is distinct. Although VAE-based anomaly detection (AD) has been extensively studied, the statistical reliability of the identified anomalies remains underexplored. Our key contribution is a method to compute $p$-values with Type I error control for anomaly regions derived from the complex operations of VAE-based AD. This is achieved by leveraging the concept of conditional selective inference (CSI) to address the issue. Thus, although scan statistics share the goal of detecting and testing anomalous regions, they fall outside the scope of our work, which specifically focuses on ensuring the statistical reliability of anomaly regions identified through VAE-based AD.
>
> > Section 4 Readability: Section 4, which appears to be the critical contribution, is dense and challenging to follow. The writing assumes familiarity with advanced concepts, placing a high cognitive load on readers who wish to fully understand the authors' approach.
>
> We thank the reviewer for their feedback on Section 4. In the updated version, we will restructure Section 4, elaborate on key steps, and include an illustrative example to improve readability.
>
> > Experimental Limitations: The experiments provide limited insights into the practical benefits and drawbacks of this approach. Since the authors make no theoretical claims about properties other than the validity of the selective p-value, and given that the motivation is high-stakes AD, the experiments are crucial.
>
> In the revised version, we will update the experimental section to include the necessary information.
> In our experiments, we set $\lambda=1.2$. While further analysis of could provide additional insights, its selection is not the primary focus of this paper. The current experiments effectively demonstrate the validity of our method in controlling the Type I error while achieving reasonable power.

---

> ### Author Response · Authors · 2024-11-23
>
> > Applicability of CSI with Alternative Region Selection: If a more sophisticated region detection approach (like those in scan statistics) were used to detect anomalous regions based on reconstruction errors, could the authors’ CSI method still be applied?
>
> To clarify, CSI is not restricted to the specific region detection method used in our paper. Theoretically, it can be applied to any approach capable of identifying the event that determines the selection of the hypothesis (anomaly region), i.e., the conditional part in Eq. (9). However, in many cases, identifying this event is challenging. To address this, we demonstrate in Section 4 that VAE-based AD can be characterized by piecewise linear operations, which makes it possible to apply CSI effectively.
>
> > Comparison to Randomization Testing: Many anomaly detection methods assess validity through randomization testing (which could be performed here, given the authors’ parametric assumptions under H0) by generating samples of data under H0, computing the test statistic in these null data samples, and comparing the test statistic in the original data to those from the null data to compute its empirical p-value.
>
> In the setting primarily focused on in this paper, the hypothesis is derived from the data through VAE-based anomaly detection, and its selection bias may influence the randomization process used to compute the $p$-value. As a result, it is not immediately clear that randomization is valid for controlling the Type I error in this situation. In contrast, CSI can provide a $p$-value with a  guarantee of Type I error control. Additionally, CSI offers the valid $p$-value without the need for approximation, whereas randomization testing requires such approximations.

---

### Official Review · Reviewer_teWT · 2024-11-04

**Soundness:** 2
**Presentation:** 2
**Contribution:** 3
**Rating:** 3
**Confidence:** 4

**Summary:**

This work presents an interesting statistical test for anomaly detection (AD) based on VAE. First of all, the authors introduce a test statistic for AD. Then they introduce CSI  for image dependent AD. Additionally, the authors present definitions of piecewise-assignment functions and piecewise linear functions to estimate the test statistic in CSI under VAE-based method. Experimental results also echo the theoretical analysis in some extent.

**Strengths:**

1. A novel and interesting idea on anomaly detection based on VAE.

**Weaknesses:**

1. There are several theoretical presentation errors which weakens the credibility of the theory and may confuse the readers. For example, in line 211, the marginal distribution is wrongly expressed. Besides, p in line 205 and 211 is not well defined before. Additionally, there may exist other wrong symbol problems and unclearly explained symbol problems in the manuscript.
2. Some symbols seem meaningless, for example, in Sec. 3, the authors split the input image into a signal space and a noise space. However, according to the analysis afterwards, I feel that it has nothing to do with the following descriptions.

To sum up, the weaknesses of 1 and 2 make the soundness and readability poor.

3. It is not clear that the VAE need to be trained from scratch or fine tuned from some pre-trained visual models.
4. As for experiments, the datasets used for verifying effectiveness of the proposed method are limited. Additionally, the comparison methods are also limited, old and not popular. There are too few quantitative results.

**Questions:**

See the weaknesses above.

---

> ### Author Response · Authors · 2024-11-23
>
> We thank the reviewer for your feedback.
>
> > 1. There are several theoretical presentation errors which weakens the credibility of the theory and may confuse the readers. For example, in line 211, the marginal distribution is wrongly expressed. Besides, p in line 205 and 211 is not well defined before. Additionally, there may exist other wrong symbol problems and unclearly explained symbol problems in the manuscript.
>
> We apologize for the typo in the equation the reviewer mentioned. The correct equation is:
>
> $$
> P_{H_0}(p \leq \alpha) = \sum_{A\in 2^{[n]}}P_{H_0}(A)P_{H_0}(p\leq \alpha \mid A_X = A)\leq \alpha
> $$
>
> We will thoroughly review the manuscript for any other similar errors in the presentation of the theory, including the precise definition of $p$ mentioned in lines 205 and 211, to ensure clarity and accuracy for the readers.
>
> > 2. Some symbols seem meaningless, for example, in Sec. 3, the authors split the input image into a signal space and a noise space. However, according to the analysis afterwards, I feel that it has nothing to do with the following descriptions.
>
> The assumption that the image consists of true signal, $\mathbf{s}$ and noise, $\mathbf{\epsilon}$, forms the foundation of the statistical testing introduced later.
> We aim to test whether the average signal value for each pixel differs between the normal region and the abnormal region identified by the VAE-based AD.
> This hypothesis test enables us to determine whether the identified regions are attributable to noise or true signals with control of Type I Error.
>
> > 3. It is not clear that the VAE need to be trained from scratch or fine tuned from some pre-trained visual models.
>
>
> The proposed method can successfully control Type I error, regardless of how the model is obtained. It can be applied to VAEs trained from scratch as well as those fine-tuned from pre-trained models. This is because our inference is performed during the testing phase, when a new test image is provided, independent of the training phase.
>
>
> > 4. As for experiments, the datasets used for verifying effectiveness of the proposed method are limited. Additionally, the comparison methods are also limited, old and not popular. There are too few quantitative results.
>
> We would like to emphasize that our proposed method is mathematically proven to be valid, ensuring proper control of Type I error without relying on any assumptions about sample size. This guarantees robust performance across diverse datasets, regardless of their size, as long as the underlying assumptions are satisfied. However, we acknowledge the limitation in the number of datasets and comparison methods and commit to extending the experimental evaluations in future work to include more datasets and comparison methods.

---

### Official Review · Reviewer_TZ3Q · 2024-11-04

**Soundness:** 3
**Presentation:** 3
**Contribution:** 2
**Rating:** 5
**Confidence:** 1

**Summary:**

This paper addresses an important problem: the lack of a statistical reliability test. The authors use VAEs for anomaly detection. They offer a test procedure and offer a theoretical framework to measure how reliable the anomaly detection process is.

**Strengths:**

+ Important problem to consider
+ Rigorous mathematical analysis

**Weaknesses:**

- One dataset with a few thousand images is not enough to establish that the statistical test is reliable in practice.

**Questions:**

-

---

> ### Author Response · Authors · 2024-11-23
>
> We thank the reviewer for your feedback.
>
> > One dataset with a few thousand images is not enough to establish that the statistical test is reliable in practice
>
> We would like to emphasize that our proposed method is mathematically proven to be valid, ensuring proper control of Type I error without relying on any assumptions about sample size. This guarantees robust performance across diverse datasets, regardless of their size, as long as the underlying assumptions are satisfied.

---

### Note · Authors · 2025-01-24

I have read and agree with the venue's withdrawal policy on behalf of myself and my co-authors.